

# Inferring soil salinity in a drip irrigation system from multi-configuration EMI measurements using Adaptive Markov Chain Monte Carlo

Khan Zaib Jadoon[1,2], Muhammad Umer Altaf[2,3], Matthew Francis McCabe[2], Ibrahim Hoteit[3], Nisar Muhammad[2], and Lutz Weihermüller[4]

[1]Department of the Civil Engineering, COMSATS Institute of Information Technology, Abbottabad 22060, Pakistan
[2]Water Desalination and Reuse Center, King Abdullah University of Science and Technology (KAUST), Thuwal, 23955-6900, Saudi Arabia
[3]Earth Science and Engineering, King Abdullah University of Science and Technology (KAUST), Thuwal, 23955-6900, Saudi Arabia
[4]Agrosphere (IBG-3), Institute of Bio- and Geosciences, Forschungszentrum Juelich, GmbH, 52425 Juelich, Germany

*Correspondence to:* Khan Zaib Jadoon
(jadoonkz@ciit.net.pk)

**Abstract.** A substantial interpretation of electromagnetic induction (EMI) measurements requires quantifying optimal model parameters and uncertainty of a nonlinear inverse problem. For this purpose, an adaptive Bayesian Markov chain Monte Carlo (MCMC) algorithm is used to assess multi-orientation and multi-offset EMI measurements in an agriculture field with non-saline and saline

soil. In the MCMC simulations, posterior distribution was computed using Bayes rule. The electromagnetic forward model based on the full solution of Maxwell's equations was used to simulate the apparent electrical conductivity measured with the configurations of EMI instrument, the CMD mini-Explorer. The model parameters and uncertainty for the three-layered earth model are investigated by using synthetic data. Our results show that in the scenario of non-saline soil, the parameters

of layer thickness are not well estimated as compared to layers electrical conductivity because layer thicknesses in the model exhibits a low sensitivity to the EMI measurements, and is hence difficult to resolve. Application of the proposed MCMC based inversion to the field measurements in a drip irrigation system demonstrate that the parameters of the model can be well estimated for the saline soil as compared to the non-saline soil, and provide useful insight about parameter uncertainty for

the assessment of the model outputs.





## 1 Introduction

Electromagnetic induction (EMI) with low frequency is a powerful tool to map the hydrological processes in the vadose zone due to the sensitivity to water content and soil salinity (Jadoon et al., 2015; Robinson et al., 2009). The use of EMI is largely motivated by the need of robust and compact

system design, easy to use, rapid acquisition, and capability to provide a large set of georeferenced measurements, which can be associated with the spatial variability of subsurface at the field scale (Corwin, 2008). The EMI instrument is used to measure soil apparent electrical conductivity ($EC_a$)-showing distribution of averaged electrical conductivity over a particular depth range. The depth of investigation of $EC_a$ depends on the coil spacing, the coil orientation, and the frequency of the

energizing field. Mester et al. (2011) reported that in the low induction number condition, the coil-orientation, offset, and frequency have major, moderate and minor effects on depth of penetration, respectively. Soil moisture, salinity and texture cannot be directly measured with EMI measurements. However, in non-saline soils, cation exchange capacity, soil moisture and texture are factors responsible for $EC_a$ variations (Rhoades et al., 1976; Sudduth et al., 2003). Whereas in saline soil,

the $EC_a$ measurement is generally dominated by the soil salinity, and the reason is the accumulation of more salt concentration in the topsoil due to the loss of water through evaporation (Corwin and Lesch, 2005; Ershadi et al., 2014). The success of EMI measurements to assess soil salinity depends on the establishment of site-specific petrophysical relationship to relate $EC_a$ with the soil salinity estimated by electrical conductivity of the saturated paste extract ($EC_e$) (Cook and Walker, 1992).

Several inversion algorithms have been developed for EMI measurements to improve the resolution of subsurface features and the assessment of soil properties (Hendrickx et al., 2002; Santos et al., 2010; Triantafilis and Monteiro Santos, 2013). The majority of these inversion algorithms solve 1-D earth model for electromagnetic wave propagation. The model of McNeill (1980) has been extensively used for low induction number and Maxwell's equations has been utilized for high conductive

soil ($EC_a > 100$ mS/m) where the low induction number assumption is not valid. For example, Li et al. (2013) used Geonics EM38 to measure $EC_a$ in a rice-paddy and did inversion using forward model of McNeill (1980) to estimate the variation of soil salinity in a field condition. They reported that the yield reduced by 33% in an irregular shaped patch of strong saline topsoil estimated by EMI inversion. EMI systems are sensitive to the field-specific calibration procedure, which limits to ob-

tain precise measurements of $EC_a$. However, in inversion modeling precise measurement of $EC_a$ is a prerequisite to characterize subsurface soil properties. For decades, the development and use of quantitative EMI inversions were mainly hampered by the lack of suitable calibration methods. von Hebel et al. (2014) used electrical resistivity tomography to calibrate EMI measurements before their inversion of EMI measurements to estimate three-dimensional imaging of subsurface electrical

conductivity. Recently, Jadoon et al. (2015) calibrated EMI measurements via vertical electrical conductivity profile measured by capacitance sensors in different pits and later performed inversion for calibrated multi-configuration EMI measurements to estimate the effect of soil salinity distribution



in an acacia tree farm.

Generally these inversion algorithms are robust and provide useful estimates of subsurface properties in terms of optimal model parameters, analysis of parameter uncertainty and correlation is often left unaddressed. Parameter uncertainty can be associated to the measurement errors (acquisition geometry, instrumental calibration and human error), modeling errors (assumptions in the electromagnetic forward model and petrophysical relationships), prior assumptions or constraints, parametrization, and inversion or estimation methods. Parameter uncertainty analysis can serve two main purposes: identify the model parameters of dominant importance and provide confidence in the estimated model parameters (Scharnagl et al., 2011). For instance, Minsley (2011) used synthetic data considering the characteristics of shallow ground-based EMI system, geophex GEM-2 (Huang and Won, 2003), to estimate parameters uncertainty for a three layer model via a Bayesian Markov Chain Monte Carlo (MCMC) approach. They showed that combining multiple configuration EMI measurements have significantly reduced total error, was best able to capture the shallow interface, and have reduced regions of uncertainty at depth.

In this work, an adaptive Bayesian MCMC algorithm was used for multi-orientation and multi-offset EMI measurements, in which the parameters posterior distribution was formulated using Bayes rule. The posterior distribution represents the complete solution of the Bayesian inversion problem, including prediction of optimal parameters value and the associated uncertainty. A uniform prior distributions over unknown parameters for the layered earth model was formulated using prior knowledge of parameters. EMI measurements were carried out under conditions of low and high induction number ($EC_a > 100$ mS/m), therefore the full solution of the Maxwell's equation was used as an electromagnetic forward model. Synthetic scenarios were analyzed for a three-layered earth model to evaluate the estimated parameter and uncertainty for saline and non-saline soil using the characteristics of EMI system, the CMD-Mini Explorer. Finally, field measurements of the CMD-Mini explorer were used to estimate parameter uncertainty of three-layered earth model and soil salinity distributions.

## 2 Materials and methods

### 2.1 Electromagnetic forward model

Forward EMI response for a given layered earth model is usually calculated by the McNeill (1980) model, which is created by using the commutative electrical conductivity distribution over a certain depth range, and valid under condition of low induction number. The alternative method used to calculate the forward EMI response is to solve the full solution of the Maxwell's equation for the magnetic field measured over a horizontal layered medium proposed by Keller and Frischknecht (1966) and Anderson (1979). An increased computational power made it possible to characterize subsurface by utilizing forward models based on the Maxwell's equation (Santos et al., 2010). On





one hand, the effective depth of exploration is independent of $EC_a$ in a low induction number condition, on other in high induction number condition inverse relationship was found between the depth of exploration and $EC_a$ (Callegary et al., 2007). For a combination of a vertical and horizontal dipole source-receiver with an offset $\rho$ over a multilayered earth, the electromagnetic forward model can be written as:

$$EC_a^{HCP}(x,\rho) = \frac{-4\rho}{\omega\mu_0}\mathrm{Im}\left[\int_0^\infty R_0 J_0(\rho\lambda)\lambda^2 d\lambda\right], \qquad (1)$$

$$EC_a^{VCP}(x,\rho) = \frac{-4}{\omega\mu_0}\mathrm{Im}\left[\int_0^\infty R_0 J_1(\rho\lambda)\lambda d\lambda\right]. \qquad (2)$$

In these expressions, $EC_a^{VCP}$ and $EC_a^{HCP}$ represents apparent electrical conductivity - measured in vertical and horizontal coplanar mode, $\mu_0$ represents permeability of the free space, $\lambda$ indicates the radial wave number, $J_0$ and $J_1$ corresponds to the zero-order and first-order Bessel functions, $\omega$ is angular frequency and Im shows the quadrature component. The reflection factor $R_0$ is obtained recursively, beginning with the lowest layer N+1, where $R_{N+1} = 0$ :

$$R_n(h_n,\sigma_n) = \frac{\frac{\Gamma_n - \Gamma_{n+1}}{\Gamma_n + \Gamma_{n+1}} + R_{n+1}\exp(-2\Gamma_{n+1}h_{n+1})}{1 + \frac{\Gamma_n - \Gamma_{n+1}}{\Gamma_n + \Gamma_{n+1}}R_{n+1}\exp(-2\Gamma_{n+1}h_{n+1})}, \qquad (3)$$

$$\Gamma_n = \sqrt{\lambda^2 + \omega\mu_0 j\sigma_n}, \qquad (4)$$

$\sigma_0 = 0$, $h_n$ is the height, and $\sigma_n$ is the electrical conductivity for the $n^{th}$ layer. The assumption made in this formulation is that each layer is uniform with infinite horizontal extent. The electromagnetic forward model, which is based on high induction number assumption, returned more reliable apparent electrical conductivity values than the standard sensitivity curves of McNeill (1980). EMI measurements were carried out under high induction number conditions ($EC_a > 100$ mS/m) resulting in utilization of the full solution of Maxwell's equation for forward EMI response. Lavoue et al. (2010) and Moghadas et al. (2012) reported that the area below the effective depth range of EMI also contributes to the apparent electrical conductivity. Keeping the above in consideration, the whole measured conductivity data upto 1.5 m depth was used for the calculations of reference apparent electrical conductivity (and also for calibration). Below this depth the electrical conductivity equals to that of the last year measured value, which is anticipated to be a homogeneous half-space.

### 2.2 Bayesian Inference

Bayesian inference approach is used to express the uncertainties in the system using a suitable likelihood function. Given a set of unknown parameters, the prior distributions of the given model are formulated and Bayes rule is then used by incorporating observational data to calculate posterior distribution (Arulampalam et al., 2002; Sivia, 2006). Bayesian inversion gained a lot of interests in recent years and has been applied in different applications, including climate, ocean and geophysical





modeling (Malinverno, 2002; Zedler et al., 2012; Olson et al., 2012; Altaf et al., 2014; Sraj et al., 2014).

Suppose a set of data ($\{y^i\}_{i=1}^n$) is available and assume a certain model to describe the data. Let $\alpha$ be the set of parameters defining our model, then according to Bayes rule

$$p(\alpha|\{y^i\}_{i=1}^n) \propto p(\{y^i\}_{i=1}^n|\alpha)\,p(\alpha), \tag{5}$$

where $p(\alpha)$ is the prior distribution of $\alpha$ that represents the a priori knowledge about $\alpha$, i.e. before
considering the data. $p(\{y^i\}_{i=1}^n|\alpha)$ denotes the likelihood function: the probability of acquiring the data given $\alpha$. $p(\alpha|\{y^i\}_{i=1}^n)$ is the posterior probability: the probability that $\alpha$ is true given the data ($\{y^i\}_{i=1}^n$).

Let's consider the forward model M, for the evaluation of the data as a function of the parameters such that:

$$y = M(\alpha). \tag{6}$$

Let $\epsilon$ be a random variable which represents the discrepancy between our model $M(\alpha)$, and the observations **y** as:

$$\epsilon = y - M(\alpha), \tag{7}$$

Specifically, we assume that $\epsilon$ follows a Gaussian distribution of mean zero and variance $\sigma^2$, i.e. $\epsilon \sim N(0, \sigma^2)$. The likelihood function can then be represented as

$$p(\{y^i\}_{i=1}^n|\alpha) = \prod_{i=1}^n \frac{1}{\sqrt{2\pi\sigma^2}} \exp\left(-\frac{(y_i - M_i(\alpha))^2}{2\sigma^2}\right). \tag{8}$$

The variance ($\sigma^2$) depends on the observational data $y$. Together with unknown parameters $\alpha$,
$\sigma^2$ is an additional unknown estimated parameter. Finally, the joint posterior distribution using the Bayesian inference is expressed as:

$$p(\alpha, \sigma^2|\{y^i\}_{i=1}^n) \propto \prod_{i=1}^n \frac{1}{\sqrt{2\pi\sigma^2}} \exp\left(-\frac{(y_i - M_i(\alpha))^2}{2\sigma^2}\right) p(\alpha)p(\sigma^2). \tag{9}$$

The choice of a prior is a key step in the inference process. Here, an informative uniform prior for all five (three conductivities and two thickness) parameters is assumed, with $\alpha_k$ in the range $[\alpha_k^{max} - \alpha_k^{min}]$; i.e:

$$p(\alpha_k) = \begin{cases} \frac{1}{\alpha_k^{max} - \alpha_k^{min}} & \text{for } \alpha_k^{min} < \alpha_k \leq \alpha_k^{max}, \\ 0 & \text{otherwise}, \end{cases} \tag{10}$$

The noise variance $\sigma^2$, we assume a Jeffreys prior (Sivia, 2006) given as:



$$p(\alpha_k) = \begin{cases} \frac{1}{\sigma^2} & \text{for } \sigma^2 > 0, \\ 0 & \text{otherwise}, \end{cases} \qquad (11)$$

The problem now reduces to simulate (sample) this posterior. Generally, the most appropriate computational strategy for a multidimensional parameters space is the Markov Chain Monte Carlo (MCMC) method. We have applied an adaptive Metropolis MCMC algorithm (Haario et al., 2001; Roberts and Rosenthal, 2009) to sample the posterior distribution.

## 3   Results and Discussion

### 3.1   Synthetic Data

Two set of scenarios were considered to test the MCMC approach to evaluate the estimated parameters and their uncertainty using synthetic data for CMD Mini-Explorer configurations. Figure 1 (a) and (b) shows a three-layer earth model of low and high conductivity for non-saline soil and high soil salinity, respectively. In both scenarios thicknesses for the three-layer earth model was conceptualized by a plow horizon (0.25 m thick), with an intermediate subsoil layer (0.50 m thick) and underlying consolidated layer up to 1.5 m depth. Usually the plowing horizon has less soil moisture as compared to the deeper horizon because of evaporation and infiltration processes. Therefore in the scenario of non-saline soil the plowing horizon had low electrical conductivity of 15 mS/m as compared to the intermediate and consolidated soil layers (Figure 1 (a)). Whereas in the saline soil scenario, salt accumulations on the surface of soil due to evaporation of water, as a result the electrical conductivity of plowing horizon, is considered higher 1800 mS/m as compared to the deeper layers (Figure 1 (b)). In the agricultural field, the increase in the soil salinity is generally due to the use of poor quality of water or the excessive use of fertilizers. Forward response of both scenarios was calculated in HCP and VCP via Equations (1) and (2), respectively, for EMI configuration setups using the characteristics of CMD-Mini Explorer of three receiver coils respectively placed at 0.32, 0.71 and 1.18 m distances from the receiver.

In both scenarios, six configurations, three each for HCP and VCP with different spacings were taken as an output for forward models. Let $\alpha = (\sigma_1, \sigma_2, \sigma_3, h_1, h_2)^T$ be a vector of model control parameters. $\sigma_1$, $\sigma_2$, and $\sigma_3$ are layer conductivities, and $h_1$ and $h_2$ thicknesses. Bayesian inference was used to estimate these 5 parameters that minimize the errors between observed and modeled HCP and VCP. An adaptive MCMC method was used to sample the posterior distributions and consequently update $\alpha$ distributions according to the observed data. All the results explained below are based on $10^4$ MCMC samples. Parameter range for $h_1$ and $h_2$ was fixed between $0.05 - 0.6$ m in each scenario. In the non-saline scenario, parameter range for $\sigma_1$, $\sigma_2$ and $\sigma_3$ was considered between 5-100 mS/m and the saline soil scenario range was fixed between 5-3000 mS/m. A uniform prior distribution function was considered in both scenarios.



Figure 2 (a) and (b) depicts the observed, estimated (modeled) and range of $EC_a$ picked from the chain of MCMC simulation for six configurations of synthetic case considered for non-saline and saline soil, respectively. X-axis represents VCP and HCP with three coil spacing ($\rho32$, $\rho71$, $\rho118$). In a non-saline scenario, the layer electrical conductivity increases with the depth (Figure 1 (a)), and is reflected in the observed and modeled $EC_a$ in the VCP and HCP with increasing trend for bigger spacing (Figure 2 (a)). The $EC_a$ value for the VCP and HCP with maximum spacing of 1.8 m between transmitter and receiver corresponds to deeper horizon and in the case of saline soil scenario the layer conductivity decreases (Figure 1 (b)) and as a result $EC_a$ values in VCP and HCP configuration exhibits a decreasing trend (Figure 2 (b)). The electromagnetic forward model is sensitive to high electrical conductive soil, so the modeled $EC_a$ values for the saline soil scenario matches well with the observed as compared to the non-saline scenario. Mismatch between the observed and modeled $EC_a$ values for non-saline soil is due to low sensitivity of the forward electromagnetic model to the low electrical conductivity.

Figure 3 (a) shows the true parameter values (red line), the value of the estimated parameters using MCMC simulations (blue dash line) for the non-saline soil scenario. The computed MCMC samples were used to obtain the marginalized posterior distributions based on kernel density estimation (KDE) (Parzen, 1962). The 95 percent of the KDE for each parameter is shown by the shaded gray background (Figure 3 a). The resulting marginalized posterior pdfs of the three conductivities and two thicknesses are shown in Figure 3 (b − f). The pdfs of each parameter (Figure 3 b−f) show a single peak, corresponding to the optimal parameter value. Electrical conductivities of three layers ($\sigma_1$, $\sigma_2$ and $\sigma_3$) were comparatively well estimated as compared to the layer thicknesses. Different uniform prior distribution functions were also considered for the layer thicknesses and in each MCMC simulation the model converges close to the prior instead of true layer thicknesses. It seems that the topography of the objective function is flat in the direction of layer thicknesses and do not change with the layer thickness picked in each iteration of the MCMC simulation. This suggests that the electromagnetic model is not sensitive to the layer thicknesses for the low conductive soil layer.

Figure 4 illustrates the true and estimated depth profile of electrical conductivity for saline scenario, and the KDE of the marginalized posterior distributions for the three layer conductivities ($\sigma_1$, $\sigma_2$ and $\sigma_3$) and the two layer thicknesses ($h_1$ and $h_2$). The shaded gray background shows the 95 percent of the KDE for each parameter (Figure 4 a). The vertical electrical conductivity profile was well optimized by MCMC simulation. The electrical conductivity of the top two layers were well estimated as compared to the consolidated layer with low electrical conductivity. Furthermore, in the six configurations of CMD Mini-Explorer, the HCP and VCP configuration with spacing 1.18 m are mostly sensitive to the consolidated layer and the remaining four configurations are sensitive to upper horizon. A big range of parameter space was searched by MCMC simulation (Figure 4 b − e), which illustrate parameters sensitivity to the electromagnetic model.





### 3.2 Experimental Data

Measurements were carried out in a farm, where acacia trees were irrigated with saline groundwater. The farm is located at a distance of 6 km from the Red Sea coast at Al-Qadeimah, Makkah province, Saudi Arabia. EMI measurements were carried out with the interval of 2 m over a 40 m-long transect, along which three acacia trees were irrigated using drip irrigation. At each location, EMI measurements using CMD-Mini explorer system gives six different values of apparent electrical conductivity (using two coil orientations and three offsets), each responds to different depth ranges. Ten pits were dug along the same transect and in each pit the vertical $\sigma_b$ profile was measured at 15 locations within a depth range of 0.05-1.5 m via 5TE capacitance sensors (Decagon Devices, Pullman, USA). 5TE and EMI measurements were carried out on the same day 8 hr after the drip irrigation system was stopped, so that the soil moisture concentration below the drippers be avoided, and the time be given for the reduction of soil moisture impact due to root water uptake, evaporation and infiltration (Jadoon et al., 2015).

Figure 5 shows soil electrical conductivity measured in ten pits along a transect and the modeled soil electrical conductivity as estimated by the Markov Chain Monte Carlo simulation for multi-configuration electromagnetic induction measurements. The pit locations along the transect are shown by black triangle and cubic interpolation of 150 5TE sensor measurements were used to construct the two dimensional profile of measured $\sigma$ (Figure 5 (a)). The groundwater used to irrigate the acacia trees has an electrical conductivity of 4200 mS/m. The three patterns of high electrical conductivity is due to infiltration front and soil salinity near three acacia trees. In total, 21 multi-configuration EMI measurements were performed along a transect and calibrated with in situ measurements obtained through capacitance sensors (Jadoon et al., 2015). Three-layer earth model was considered for Bayesian inference to estimate five parameters ($\sigma_1, \sigma_2, \sigma_3, h_1, h_2$) and their uncertainty based on the 15,000 MCMC samples. For all MCMC simulations, the parameter space for optimization was set relatively large, having the range of values used for low and high electrical conductivity of soil; namely, $0 < \sigma_1 < 3000$ mS/m, $0 < \sigma_2 < 3000$ mS/m, $0 < \sigma_3 < 3000$ mS/m, $0.05 < h_1 < 0.6$ m, and $0.05 < h_1 < 0.6$ m. In the depth section of soil electrical conductivity obtained by EMI MCMC simulations, the effect of infiltration patterns and the soil salinity due to the drip irrigation near three acacia trees can be observed (Figure 5 (b)). The obtained soil electrical conductivity values by MCMC simulation are in a good agreement with sensor measurements performed in pits (Figure 5 (a)).

Figure 6 (a) and (b) shows the measured, estimated (modeled) and range of $EC_a$ picked from the chain of MCMC simulation for six multi-configuration of CMD-Mini Explorer for non-siline and saline soil, respectively. Three coil spacing for each VCP and HCP is represented on x-axis. EMI measurement is shown for non-saline and saline soil is at the location 4 and 9 of the pit (Figure 5 (a)), respectively. The soil was completely dry for non-saline soil as no irrigation was applied, whereas in the case of saline soil the moisture in the soil was in the range of 0.005-0.19 at the time of EMI



and sensor measurements. In non-saline soil, the measured six $EC_a$ values are in the range of 5-60 mS/m and the modeled $EC_a$ value are in the range of 23-38 mS/m Figure (6 (a)). The range of $EC_a$ picked from the last 10,000 MCMC simulation is in the range of 0-75 mS/m. As we observed in the synthetic non-saline soil scenario that the electromagnetic forward model was not sensitive to the low electrical conductive soil similarly the fit between the measured and modeled $EC_a$ is not in good agreement with the real measurements (Figure 6 (a)). Furthermore, the misfit may be due to the large search parameter space in the MCMC simulations. In the case of saline soil, the electrical conductivity of the top 50 cm soil is high due to the saline infiltration and soil salinity. This effect can be seen in the decreasing trend of the measured $EC_a$ for the VCP and HCP measurements with bigger coil spacing (Figure 6 (b)). The measured and modeled $EC_a$ are in good agrement and this is due to the sensitivity of the electromagnetic forward model to high electrical conductive soil.

Figure 7 plots the vertical profile of electrical conductivity for non-saline soil measured by capacitance sensors (red line), the value of the estimated parameters using the MCMC simulations (blue dash line), and the KDE of the marginalized posterior distributions for the three layer conductivities and the two layer thicknesses. CMD-Mini Explorer measurements at the pit 4 for non-saline soil was used for the analysis. In Figure 7 (a), the shaded area shows the 95% KDE distribution limits, the measured vertical profile of soil electrical conductivity fall within the shaded area in the top depth 0-0.7 m and below this depth modeled soil electrical conductivity is over estimated. The mismatch between the measured and modeled $EC_a$ for the maximum coil separation $H\rho118$ and $V\rho118$ is the cause of over estimation of modeled soil electrical conductivity. The marginalized posterior pdfs of the three conductivities and two thicknesses are shown in Figure 7 (b − f). The pdfs of each parameter (Figure 7 b−f) exhibit a single peak and corresponds to the optimal parameter value. The peak of the $\sigma_3$ is flat between 30-38 mS/m and seems the topography of the objective function do not change within this range of conductivity in each iteration of the MCMC simulation.

Finally, Figure 8 plots the vertical profile of electrical conductivity for saline soil measured by capacitance sensors (red line), the value of the estimated parameters using the MCMC simulations (blue dash line), and the KDE of the marginalized posterior distributions for the three layer conductivities and the two layer thicknesses. CMD-Mini Explorer measurements at the pit 9 for saline soil was used for the analysis. The shaded area in Figure 8 (a) plots the 95% KDE distribution limits, and the whole measured vertical profile of soil electrical conductivity fall within the shaded area. This suggests that the electrical conductivity is well estimated. The marginalized posterior pdfs of the three conductivities and two thicknesses as shown in Figure 8 (b − f), exhibit a single peak for each parameter except layer thickness $h_2$ which is flat which shows that the measured data were not useful to refine our prior knowledge for $h_2$. The posterior pdfs of first two conductivities ($\sigma_1$ and $\sigma_2$) and layer thickness $h_1$ appear to be a precise Gaussian shape with a clear Maximum A Posteriori (MAP) values. For conductivity parameter $\sigma_3$, we notice a posterior with a well defined peak but no clear pdf shape.



Conventional estimation of a single best-fit model with linear uncertainty usually does not trace ambiguity in the models, and may lead to a misguiding or imprecise interpretation. For instance, Jadoon et al. (2015) used global optimization algorithm to estimate soil salinity by using multi-configuration EMI measurements without estimating uncertainties in the model parameters. A comprehensive strategy is thus required for the assessment of non-uniqueness and uncertainty in the model parameters. This research has attempted to evaluate model parameters and their uncertainties using the Bayesian inference framework for both synthetic and ground-based EMI field measurements to estimate the soil salinity in a drip irrigation system. Such analysis helps to provide insight about parameters estimate and uncertainties. The synthetic and the field measurements show that the electromagnetic forward model used for the CMD-mini explorer measurements is less sensitive to the layer thicknesses. Furthermore, the model parameters for the saline soil can be well estimated as compared to the case of non-saline soil. Future research will focus to implement the Bayesian inference approach on time-lapse EMI measurements in different agricultural fields to monitor the soil dynamics, estimate the model parameters and their uncertainties.

## 4 Conclusion

An adaptive Bayesian MCMC algorithm has been introduced for the model assessment and uncertainty analysis of multi-orientation and multi-offset EMI measurements. The algorithm has been tested for CMD-Mini Explorer with both synthetic and field measurements conducted in an agriculture field over a non-saline and saline soil. Using Bayesian inference, marginalized posterior pdfs were computed for three subsurface electrical conductivities ($\sigma_1$, $\sigma_2$, and $\sigma_3$) and two layer thicknesses ($h_1$ and $h_2$) using MCMC. To the best of the authors' knowledge, this is the first study in which the MCMC technique is incorporated for both the saline and non-saline soils for realistic low frequency EMI measurements.

The experimental results showed that the MCMC simulations can improve the reliability of the electromagnetic forward model to estimate the subsurface electrical conductivity profiles. Analysis shows that the electromagnetic forward model is less sensitive to the non-saline soil as compared to the saline soil. The proposed approach is flexible and can be implemented for various low-frequency ground-based EMI system and can provide subsurface electrical conductivity distribution and uncertainty of model parameters.

*Acknowledgements.* This research was funded by the Water Desalination and Reuse Center, King Abdullah University of Science and Technology (KAUST), Saudi Arabia.



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





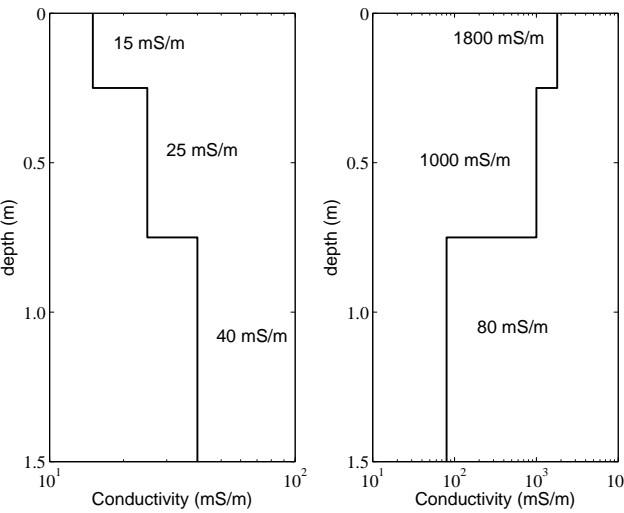

**Fig. 1.** Three-layer synthetic earth model of electrical conductivity for (a) non-saline soil and (b) saline soil in the top horizon.

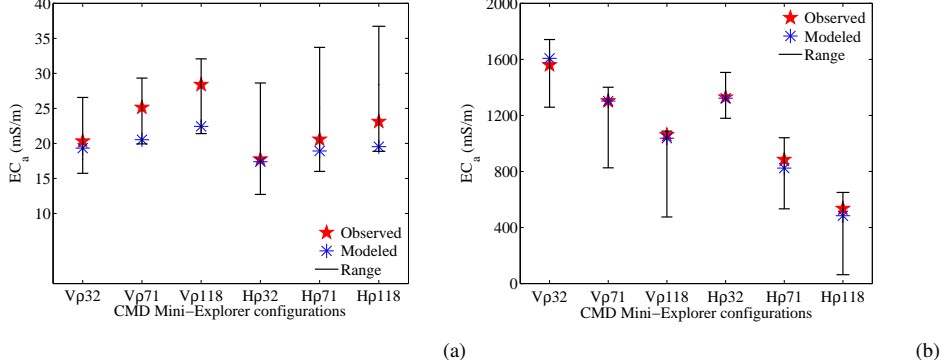

**Fig. 2.** Observed electrical conductivity obtained from the forward response of the six different configuration of CMD-Mini Explorer (red star), estimated (modeled) earth electrical conductivity (blue asterisk) and the range of $EC_a$ simulated by MCMC for (a) non-saline and (b) saline soil scenarios.





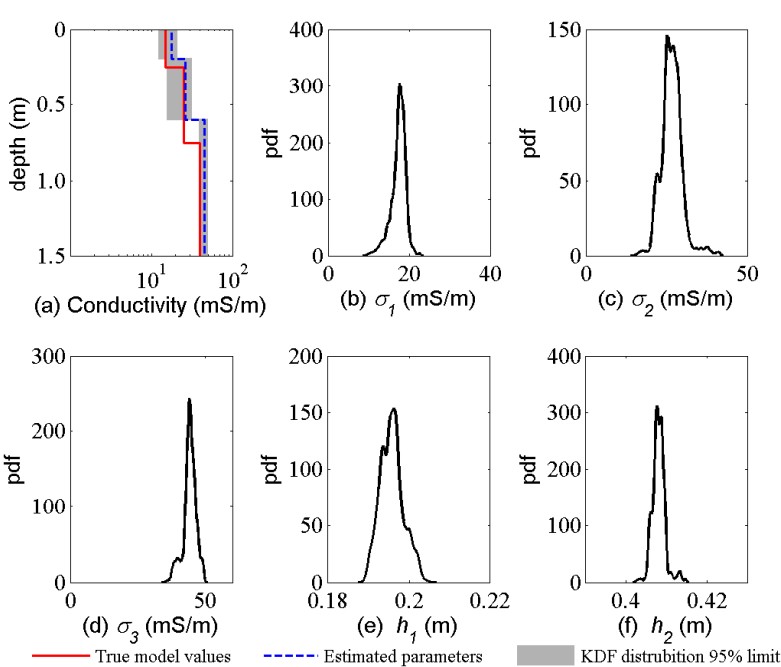

**Fig. 3.** Summary of the MCMC simulation for the synthetic three layer earth model of non-saline soil. (a) True (red line) and estimated parameter (blue dash-line) for the vertical electrical conductivity profile, and the gray background with the 95 percent confidence interval of kernel distribution estimation (KDE). (b − f) show the KDE of the marginalized posterior distributions for the three layer conductivities ($\sigma_1$, $\sigma_2$ and $\sigma_3$) and two layer thicknesses ($h_1$ and $h_2$).





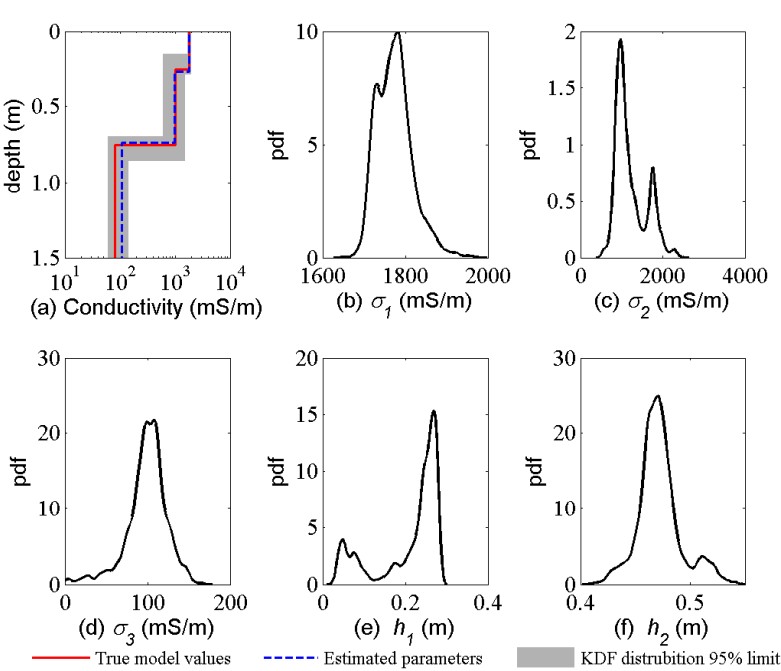

**Fig. 4.** Summary of the MCMC simulation for the synthetic three layer earth model of saline soil. (a) True (red line) and estimated parameter (blue dash-line) for the vertical electrical conductivity profile, and the gray background with the 95 percent confidence interval of kernel distribution estimation (KDE). (b − f) show the KDE of the marginalized posterior distributions for the three layer conductivities ($\sigma_1$, $\sigma_2$ and $\sigma_3$) and two layer thicknesses ($h_1$ and $h_2$).





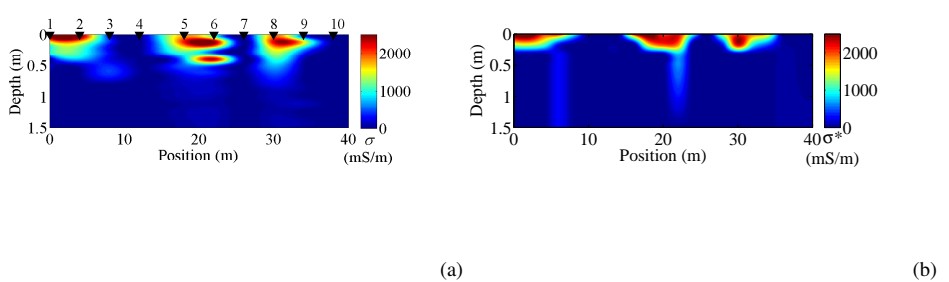

(a)                 (b)

**Fig. 5.** (a) electrical conductivity (mS/m) measured by the 5TE capacitance sensors from 10 soil pits along transect and the location of the soil pits is indicated by black triangles (Jadoon et al., 2015), (b) the soil electrical conductivity obtained by using Markov Chain Monte Carlo simulation for multi-configuration electromagnetic induction measurements.

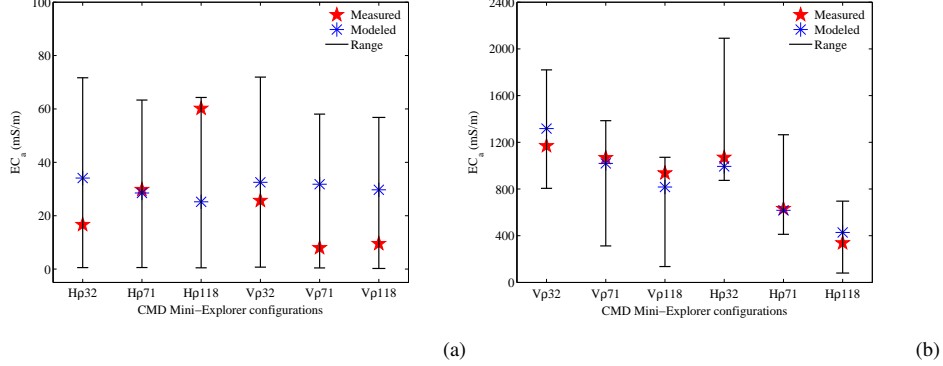

(a)                 (b)

**Fig. 6.** Measured six different configuration of CMD-Mini Explorer (red star), estimated (modeled) earth electrical conductivity (blue asterisk) and the range of $EC_a$ simulated by MCMC for (a) non-saline soil at pit 4 and (b) saline soil at pit 9 location.





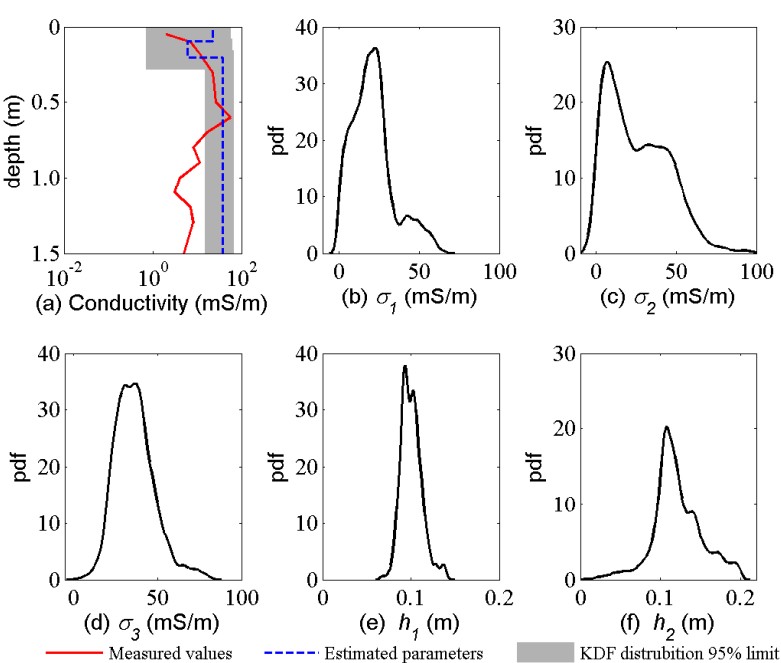

**Fig. 7.** Summary of the MCMC simulation for three-layer earth model by considering CMD-Mini explorer measurement over a non-saline soil. (a) True (red line) and estimated parameter (blue dash-line) for the vertical electrical conductivity profile, and the gray background with the 95 percent confidence interval of kernel distribution estimation (KDE). (b − f) show the KDE of the marginalized posterior distributions for the three layer conductivities ($\sigma_1$, $\sigma_2$ and $\sigma_3$) and two layer thicknesses ($h_1$ and $h_2$).





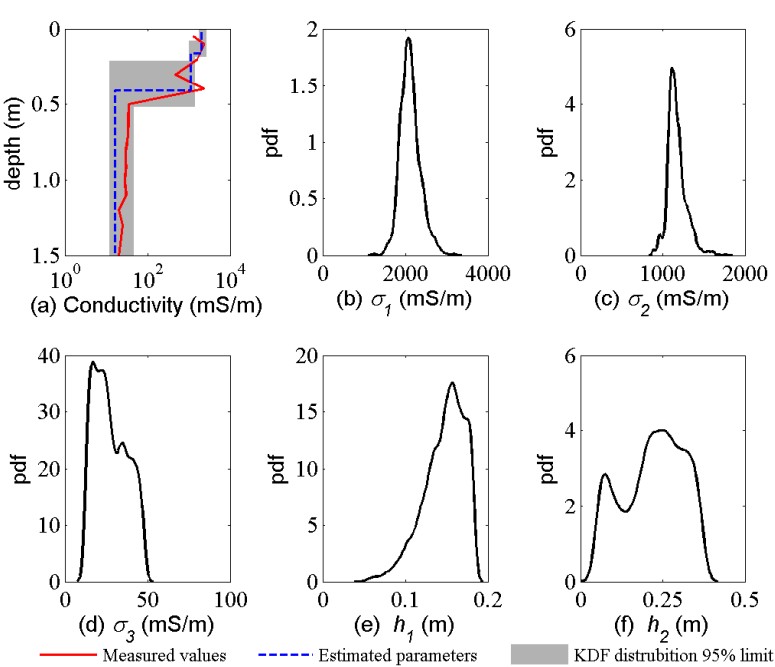

**Fig. 8.** Summary of the MCMC simulation for three-layer earth model by considering CMD-Mini explorer measurement over a saline soil. (a) True (red line) and estimated parameter (blue dash-line) for the vertical electrical conductivity profile, and the gray background with the 95 percent confidence interval of kernel distribution estimation (KDE). (b − f) show the KDE of the marginalized posterior distributions for the three layer conductivities ($\sigma_1$, $\sigma_2$ and $\sigma_3$) and two layer thicknesses ($h_1$ and $h_2$).