# Peer review of "Inferring soil salinity in a drip irrigation system from multi-configuration EMI measurements using Adaptive Markov Chain Monte Carlo"

_Hydrology and Earth System Sciences, 2016_

## Referee Comment (RC1) · Anonymous Referee #1 · 7 Sep 2016

The paper by Jadoon et al. addresses the estimation of parameter error for inversion of electromagnetic induction measurements, using a Bayesian framework. Overall the inversion approach and the Bayesian procedure for parameter error estimation have valid scientific merit. Nevertheless, I have some concerns on the experimental dataset that was used. The structure of the paper needs moderate revisions.

Specific comments Title. The paper never attempts to calibrate the ECa readings to actual salinity estimations. Therefore the title is not reflecting the contents of the paper. ECa does not equal to salinity. The interpretation of ECa is much more complex. The experimental data of the paper deals with a highly conductive medium (wet & saline soil) and a non-conductive one (dry soil). The results should be therefore discussed in

this light.

L67-78. This is a summary of what's done in the paper. I would rewrite this section featuring what the objectives of the paper are. (If you want to guide the reader through the paper by explaining the workflow, then do it at the beginning of the Materials and methods section.

L79. Start the Materials and methods section by describing the synthetic and experimental data. Currently such descriptions are in the Results and Discussion session: they do not belong there. L98-99 "The assumption made in this formulation is that each layer is uniform with infinite horizontal extent." It would not hurt to know a little more about this assumption. L99-101- "The electromagnetic forward model, which is based on high induction number assumption, returned more reliable apparent electrical conductivity values than the standard sensitivity curves of McNeill (1980)." This should be moved up (L 82?) and rephrased as: "preliminary analyses indicated that. . ." L103-108 "Lavoue et al. (2010) and Moghadas et al. (2012) . . . to be a homogeneous half-space." These lines seem a little out of place here. Maybe you should move them to the section where the experimental data is described.

L118 I think you should explain Eq (5) in words so to warrant faster understanding.

L134 "Here, an informative uniform prior for all five (three conductivities and two thickness) parameters" Describe the parameters earlier on. L135. Awkward wording: maybe a verb is missing? L136. "The problem now reduces to simulate (sample) this posterior." Awkward phrasing: reword the sentence.

L142-167. This part belongs in the Materials and methods section. L185. pdf (?): define at first appearance

L204. Experimental data. This is the section that needs the most re-writing. Large portions of this belong in the Materials and methods section. Specific notes. -Pullman is in the state of Washington. (there are other 4 cities with the same name in the USA)

- Could not understand the sentence "5TE and EMI measurements were carried out on the same day 8 hr after the drip irrigation system was stopped, so that the soil moisture concentration below the drippers be avoided, and the time be given for the reduction of soil moisture impact due to root water uptake, evaporation and infiltration (Jadoon et al., 2015)."

L239 here's my only methodological issue with this paper. You use as non-conductive scenario a soil that is completely dry. (By the way, what are the salinity values measured at this site? E.g., the conductivity of the saturated paste extract?). In their protocols for use of apparent electrical conductivity measurements in agriculture, Corwin and Lesch state that the soil volumetric water content should be at least 50% of the value at field capacity (ideally between 70% and field capacity. Otherwise, the liquid pathways of electrical conductivity through the soils would be interrupted, unpredictably increasing the resistivity of the soil. This is very likely reason why your results on the non-conductive scenario are not encouraging. My criticism is the following: with one scenario where ECa is known not to be reliable, is the other scenario (highly conductive medium) enough to provide context to your data analyses? I fear not. I think this paper would make much better of a point if other scenarios (e.g., increasing water contents?) were presented. See: Corwin, D.L., and S.M. Lesch. 2013. Protocols and guidelines for field-scale measurement of soil salinity distribution with ECa-directed soil sampling. J. Environ. Eng. Geophysics 18(1):1-25. and: Corwin, D.L., and S.M. Lesch. 2005b. Characterizing soil spatial variability with apparent soil electrical conductivity: I. Survey protocols. Comput. Electron. Agric. 46(1-3):103-134.

L283-291 this section should be rephrased and moved to the conclusion L292-3006 generally, this section is not a conclusion but a summary.

---

## Author Comment (AC1) · 21 Nov 2016

Dear Editor and Reviewers.

We appreciate your constructive comments and thank you for the time spent on reviewing our work. Your comments and suggestions have certainly improved the scope and focus of the manuscript. Please find below our detailed answers to the reviewer comments.

The paper by Jadoon et al. addresses the estimation of parameter error for inversion of electromagnetic induction measurements, using a Bayesian framework. Overall the inversion approach and the Bayesian procedure for parameter error estimation have

valid scientific merit. Nevertheless, I have some concerns on the experimental dataset that was used. The structure of the paper needs moderate revisions.

1.1: Specific comments Title. The paper never attempts to calibrate the ECa readings to actual salinity estimations. Therefore the title is not reflecting the contents of the paper. ECa does not equal to salinity. The interpretation of ECa is much more complex. The experimental data of the paper deals with a highly conductive medium (wet & saline soil) and a non-conductive one (dry soil). The results should be therefore discussed in this light.

Reply 1.1: Indeed, in this paper a relationship was not developed to calibrate the ECa readings to actual salinity. In the revised manuscript a relationship that estimates soil salinity from ECa, which was previously established in Jadoon et al. WRR (2015) for the same site, will be included. The interpretation of ECa is complex. Nevertheless, in the saline soil the ECa measurement is generally dominated by the soil salinity. A figure related to the soil salinity will be included in the revised manuscript (see Figure 9 below) and results will be discussed in the context of ECa and later soil salinity. For example:

"Figure 9 shows the spatial distribution of soil salinity estimated from EMI measurement using Bayesian MCMC simulations. Soil salinity ECe is related to bulk electrical conductivity $\sigma$b via a linear relationship (ECe = 13.74 $\sigma$b + 0.001) established by Jadoon et al. (2015) for the same site. Infiltration front and high soil salinity ranges from 0.01 to 0.5 m at three locations where Acacia trees are irrigated with brackish water. Results show that the Bayesian inversion of multi-configuration EMI measurement permits the estimation of soil salinity caused by the brackish water infiltration. In the field, Acacia roots were concentrated in the top 70 cm of soil and the low soil salinity below 30 cm shows that Acacia are capable of extracting salt solutions and reduce subsoil salinity. Furthermore, the water content in the top soil was 53% of the field capacity".

1.2: L67-78. This is a summary of what's done in the paper. I would rewrite this section

featuring what the objectives of the paper are. (If you want to guide the reader through the paper by explaining the workflow, then do it at the beginning of the Materials and methods section.

Reply 1.2: The final paragraph of the introduction will be rewritten as below:

"Conventional estimation of a single best-fit model with linear uncertainty usually does not trace ambiguity in the models, and may lead to a misguiding or imprecise interpretation. In this work, an adaptive Bayesian MCMC algorithm was used for multi-orientation and multi-offset EMI measurements, in which the parameters posterior distribution represents the complete solution of the Bayesian inversion problem, including prediction of optimal parameters value and the associated uncertainty. Synthetic scenarios were analyzed for a three-layered earth model to evaluate the estimated parameter and uncertainty for saline and non-saline soil using the characteristics of the CMD-Mini Explorer EMI system. Furthermore, field measurements of the CMD-Mini explorer were used to estimate parameter uncertainty in the three-layered earth model and soil salinity distributions in an agricultural field irrigated with drip irrigation system."

1.3: L79. Start the Materials and methods section by describing the synthetic and experimental data. Currently such descriptions are in the Results and Discussion session: they do not belong there. L98-99 "The assumption made in this formulation is that each layer is uniform with infinite horizontal extent." It would not hurt to know a little more about this assumption.

Reply 1.3: Following the suggestion, in the revised manuscript the setup of synthetic and field measurements will be incorporated in the Material and Methods section with the subtitle: "Synthetic and Field measurements"

1.4: L99-101- "The electromagnetic forward model, which is based on high induction number assumption, returned more reliable apparent electrical conductivity values than the standard sensitivity curves of McNeill (1980)." This should be moved up (L 82?) and rephrased as: "preliminary analyses indicated that:" L103-108 "Lavoue et al. (2010)

and Moghadas et al. (2012) : : : to be a homogeneous half-space." These lines seem a little out of place here. Maybe you should move them to the section where the experimental data is described.

Reply 1.4: The sentence will be moved and rephrased as suggested.

1.5: L118 I think you should explain Eq (5) in words so to warrant faster understanding.

Reply 1.5: Following text will be incorporated in the revised manuscript: The equation (5) refers to Bayes law which describes the probability of an event, based on conditions that might be related to the event. One of the many applications of Bayes theorem is Bayesian inference.

1.6: L134 "Here, an informative uniform prior for all five (three conductivities and two thickness) parameters" Describe the parameters earlier on. L135. Awkward wording: maybe a verb is missing? L136. "The problem now reduces to simulate (sample) this posterior." Awkward phrasing: reword the sentence.

Reply 1.6: Sentence will be rephrased as: "The sentence will be rephrased as suggested."

1.7: L142-167. This part belongs in the Materials and methods section. L185. pdf (?): define at first appearance

Reply 1.7: Both paragraphs will be moved to the Materials and methods section.

1.8: L204. Experimental data. This is the section that needs the most re-writing. Large portions of this belong in the Materials and methods section. Specific notes. –Pullman is in the state of Washington. (there are other 4 cities with the same name in the USA) - Could not understand the sentence "5TE and EMI measurements were carried out on the same day 8 hr after the drip irrigation system was stopped, so that the soil moisture concentration below the drippers be avoided, and the time be given for the reduction of soil moisture impact due to root water uptake, evaporation and infiltration (Jadoon et al., 2015)."

Reply 1.8: The entire paragraph will be moved to the Material and methods section. The sentence is rephrased as below:

"EMI and 5TE measurements were performed 8 h after the drip irrigation system was stopped, to avoid concentrated patterns of soil moisture below the drippers and to allow some time to for evaporative losses, root water uptake and infiltration to reduce the soil moisture impact (Jadoon et al., 2015)."

1.9: L239 here's my only methodological issue with this paper. You use as non-conductive scenario a soil that is completely dry. (By the way, what are the salinity values measured at this site? E.g., the conductivity of the saturated paste extract?). In their protocols for use of apparent electrical conductivity measurements in agriculture, Corwin and Lesch state that the soil volumetric water content should be at least 50% of the value at field capacity (ideally between 70% and field capacity. Otherwise, the liquid pathways of electrical conductivity through the soils would be interrupted, unpredictably increasing the resistivity of the soil. This is very likely reason why your results on the non-conductive scenario are not encouraging. My criticism is the following: with one scenario where ECa is known not to be reliable, is the other scenario (highly conductive medium) enough to provide context to your data analyses? I fear not. I think this paper would make much better of a point if other scenarios (e.g., increasing water contents?) were presented. See: Corwin, D.L., and S.M. Lesch. 2013. Protocols and guidelines for field-scale measurement of soil salinity distribution with ECa-directed soil sampling. J. Environ. Eng. Geophysics 18(1):1-25. and: Corwin, D.L., and S.M. Lesch. 2005b. Characterizing soil spatial variability with apparent soil electrical conductivity: I. Survey protocols. Comput. Electron. Agric. 46(1-3):103-134.

Reply 1.9: We thank the reviewer for highlighting this important issue. For the same site, Jadoon et al. (2015) reported a relationship to relate bulk electrical conductivity to the soil salinity (i.e., the conductivity of the saturated paste extract). Observed soil salinity range between 3-185 dS/m). As discussed earlier (Reply 1.1), in the revised manuscript same relationship will be used to estimate the soil salinity. Text and Figure

9 will be incorporated to show the soil salinity distribution.

Synthetic and field measurements were analysed to test the performance of the electromagnetic forward model in conductive and non-conductive soil, and retrieve soil salinity using Bayesian inversion. In the case of synthetic scenarios, EMI data was generated using electromagnetic forward model and Bayesian inversion was used to estimate five parameters (three layer electrical conductivities and two layer thicknesses). Result shows that the electromagnetic forward model is not sensitive to the non-conductive soil. Similarly, Minsley (2011) used synthetic data considering the characteristics of shallow ground-based EMI system, geophex GEM-2 and reported that the electromagnetic forward model is less sensitive to the non-conductive soil. Indeed, in the agriculture field the soil electrical conductivity decreases if the soil water content is below 50% of the field capacity, which may cause the less encouraging results for the non-conductive soils. This issue will be highlighted in the manuscript and the references of Corwin, D.L., and S.M. Lesch. 2005 and 2013 will be incorporate.

In the synthetic scenario of non-saline soil, the increasing trend of soil moisture with depth has been analysed (Figure 1a). While certainly very interesting, undertaking time-lapse EMI measurement with varying soil moisture dynamics is beyond the scope of this current contribution.

1.10: L283-291 this section should be rephrased and moved to the conclusion L292-3006 generally, this section is not a conclusion but a summary. Reply 1.3: Sentences will be rephrased and conclusion will been improved as suggested.

References: Jadoon K. Z., Moghadas D., Jadoon A., Missimer T., Al-Mashharawi S., and McCabe M. F., 2015. Estimation of soil salinity in a drip irrigation system by using joint inversion of multi-coil electromagnetic induction measurements, Water Resources Research, volume 51, issue 5, page 3490-3504 DOI: 10.1002/2014WR016245

Please also note the supplement to this comment:

http://www.hydrol-earth-syst-sci-discuss.net/hess-2016-299/hess-2016-299-AC1-supplement.pdf

---

## Author Response (AR2)

Dear Editor and Reviewer,

We appreciate your constructive comments and thank you for the time spent on reviewing our revised manuscript. Your comments and suggestions have certainly improved the quality of the manuscript, which we greatly acknowledge. Please find below our detailed answers to the reviewer comments.

The manuscript have been revised by one reviewer. In their replies, the authors seems confident to be able to address the main issues raised during the reviewer process. After my own re-reading of the manuscript, I also have the following comments:

1.1- In the manuscript (abstract and introduction) the authors talk about parameters uncertainty and model uncertainty. Indeed in their application they only consider "parameter uncertainty" since only one model is considered. This point should be clarified in the manuscript. For example in the abstract the sentence "The model parameters and uncertainty " is misleading. I guess the authors mean "uncertainty in model parameters".

Reply 1.1:
Only one model is considered in this study and the uncertainty in model parameters is investigated. In the abstract and introduction, the wording has been changed to "uncertainty in the model parameters"

1.2- Introduction: page 2 last sentence "Generally ....analysis of parameter uncertainty and correlation is often left unaddressed". This statement is false. In the literature (and in particular in hydrology – hydrogeology) there is a large number of studies dealing with parameters uncertainty (starting at least from the '80).

Reply 1.2:
This is correct. Indeed, in hydrology and hydrogeology a large number of studies have addressed parameters uncertainty. However, in the case of EMI data analysis uncertainty in model parameters is often neglected, and this was the intent of the sentence here. We have addressed this by making a more specific statement.

1.3- Page 3, line 7 from the bottom "... to solve the full solution..."please rephrase

Reply 1.3:
In the revised manuscript the sentence has been rephrase as below:
*"The alternative method used to calculate the forward EMI response is to solve the Maxwell's equation for the magnetic field measured over a horizontal layered medium, as proposed by Keller and Frischknecht (1966) and Anderson (1979)."*

1.4- Page 4, eq (1). Symbol x not defined.
Reply 1.4:
Symbol "x" has now been defined as depth of the layer.

1.6- Page 4, line 6 from the bottom "….the prior distributions of a given model….". This sentence is not clear. Do you the authors mean the prior distributions of parameters a given model?

Reply 1.6:
The sentence was unclear and is now modified as below:
*"Given a set of unknown parameters, the prior distributions of the model parameters are formulated and Bayes rule is then used to calculate posterior distribution conditioned on available observations (Arulampalam et al., 2002; Sivia, 2006)."*

1.7- Page 5, symbol y two lines after eq (6) should not be in bold

Reply 1.7:
Bold option in the font settings has been deactivated for the symbol "y"

1.8- Eq (11). I am confused by this equation for the pdf of sigma. Sigma is defined as a positive (unbounded) variable. Therefore the integral of (11) between 0 and Infinity must be equal to 1. This is not true according to (11).

Reply 1.8:
Using the inverse gamma distribution, One gets

$$P(\sigma^2|\alpha, \beta) \propto (\sigma^2)^{-\alpha-1}\exp(-\frac{\beta}{\sigma^2})$$

Now taking $\beta \rightarrow 0$ and $\alpha \rightarrow 0$ then the inverse gamma will approach the Jeffrey's prior. This distribution is called "uninformative" because it is a proper approximation to the Jeffreys prior

$$P(\sigma^2) \propto \frac{1}{\sigma^2}$$

which is uninformative for scale parameter, because this prior is the only one which remains invariant under a change of scale (note that the approximation is not invariant). This has a indefinite integral of $\log(\sigma^2)$ which shows that it is improper if the range of $\sigma^2$ includes either 0 or ∞. We don't observe infinite value for variance, and if the observed variance is zero, we have perfect data. So we can set a lower limit equal to L>0, and upper limit equal U<∞, and our distribution becomes proper. A better non-informative distribution can be chosen as the upper and lower limits L and U in the Jeffreys prior. Usually the limits can be set fairly easily with a bit of thought to what $\sigma^2$ actually means in the real world.

1.9- Page 6. The authors introduce the parameters to be estimated. These include (in the 3 layer system) 3 conductivities and only 2 layer thickness. Why the thickness of the third layer is not estimated?

Reply 1.9:
For the inversion algorithms of EMI data the thickness of the last layer is assumed to be infinite as the response of EMI signal is weak for deeper depths. This kind of approach is generally applied in EMI inversion studies such as those of Lavoué et al. (2010).

1.10- Page 8. Parameters sigma and sigma_b are not defined

Reply 1.10:
Sigma and sigma_b are now defined in the revised manuscript.

1.11- Page 8, line 8 from the bottom. Typo: replace siline by saline

Reply 1.11:
Typo mistake has been corrected.

1.12- Please consider to use the help of an expert in English usage and grammar to revise the manuscript

Reply 1.12:
The manuscript is now revised for English language by a native speaker.

1.13- If the authors decide to revise and resubmit the manuscript to HESS they should carefully address these issues together with the main criticisms pointed out by the reviewer.

The main criticisms pointed out by the reviewer were:
1.13a- "My only methodological issue with this paper. You use as non-conductive scenario a soil that is completely dry. (By the way, what are the salinity values measured at this site? E.g., the conductivity of the saturated paste extract?).
Reply 1.13a:
We thank the reviewer for highlighting this important issue. As reported in the last revision that for the same site, Jadoon et al. (2015) proposed a relationship to relate bulk electrical conductivity to the soil salinity (i.e., the conductivity of the saturated paste extract). Observed soil salinity range between 3-185 dS/m. In the last revision, the same relationship was used to estimate the soil salinity. Additional text and Figure 9 were incorporated to show the soil salinity distribution.

1.13b- In their protocols for use of apparent electrical conductivity measurements in agriculture, Corwin and Lesch state that the soil volumetric water content should be at least 50% of the value at field capacity (ideally between 70% and field capacity. Otherwise, the liquid pathways of electrical conductivity through the soils would be interrupted, unpredictably increasing the resistivity of the soil. This is very likely reason why your results on the non-conductive scenario are not encouraging.

Reply1.13b:
Indeed, in agriculture fields soil apparent electrical conductivity decreases if the soil volumetric water content decreases below 50% of the value at field capacity, whereby the process described by Corwin and Lesch greatly depends on soil textural distribution. Additionally, if non-saline water is used for irrigation the soil water content dominates the EC readings. In our study, on the other hand, saline groundwater was used to irrigate the Acacia tree and salt starts to accumulate in the top soil when volumetric soil water content decreases. Therefore, the accumulated salt has a dominating effect on soil apparent electrical conductivity and also the hydroscopic salts will provide liquid pathways even in this dry environment. Furthermore, in the field, non-conductive

soil were at locations between the Acacia trees where drip irrigation system was not used to irrigate the farm so there was no change in the soil water content for the non-conductive soil.

1.13c- My criticism is the following: with one scenario where ECa is known not to be reliable, is the other scenario (highly conductive medium) enough to provide context to your data analyses? I fear not. I think this paper would make much better of a point if other scenarios (e.g., increasing water contents?) were presented. See: Corwin, D.L., and S.M. Lesch. 2013. Protocols and guidelines for field-scale measurement of soil salinity distribution with ECa-directed soil sampling.
J. Environ. Eng. Geophysics 18(1):1-25. and: Corwin, D.L., and S.M. Lesch. 2005b. Characterizing soil spatial variability with apparent soil electrical conductivity: I. Survey protocols. Comput. Electron. Agric. 46(1-3):103-134."

Reply 1.13c:
In general, the protocols developed by Corwin and Lesch are mainly based on observations in combination with theoretical knowledge of current flow in porous media. Even if these protocols are of high value they are restricted to observational findings in specific environments. Therefore, we believe that a full physical description using synthetic scenarios in combination with observations will increase our understanding of EMI sensing for areas such as those being analysed here.
As such, the synthetic scenarios were analysed to test the performance of the electromagnetic forward model in conductive and non-conductive soil, with uncertainty in model parameters estimated using Bayesian inversion. In the case of synthetic scenario of non-saline soil, the increasing trend of soil moisture with depth has been analysed (Figure 1a). Results show that the electromagnetic forward model is not sensitive to the non-conductive soil. Previous studies have reported similar results for different EMI systems. For instance, Minsley (2011) used synthetic data considering the characteristics of shallow ground-based EMI system (Geophex GEM-2) and reported that the electromagnetic forward model is less sensitive to the non-conductive soil. While certainly very interesting, undertaking time-lapse EMI measurement with varying soil moisture dynamics is beyond the scope of this current contribution.

[revised manuscript text omitted]

---

## Author Response (AR3)

Dear Editor and Reviewer,

We appreciate your constructive comments and thank you for accepting our manuscript with minor revision. Please find below detailed reply to your comments.

1.1 The revision of hess-2016-299 is a good improvement compared to the original submission. The materials and methods are now clear. Conclusions are adequate and supported by the results. All responses to my comments and to the Editors's comments are adequate.

Reply 1.1:

We acknowledged Editor and reviewers comments on our previously revised manuscript, which improved the quality of our manuscript.

1.2 I have just one note. In my original report I commented:

1.9: L239 here's my only methodological issue with this paper. You use as nonconductive scenario a soil that is completely dry. (By the way, what are the salinity values measured at this site? E.g., the conductivity of the saturated paste extract?). In their protocols for use of apparent electrical conductivity measurements in agriculture, Corwin and Lesch state that the soil volumetric water content should be at least 50% of the value at field capacity (ideally between 70% and field capacity. Otherwise, the liquid pathways of electrical conductivity through the soils would be interrupted, unpredictably increasing the resistivity of the soil. This is very likely reason why your results on the non-conductive scenario are not encouraging. My criticism is the following: with one scenario where ECa is known not to be reliable, is the other scenario (highly conductive medium) enough to provide context to your data analyses? I fear not. I think this paper would make much better of a point if other scenarios (e.g., increasing water contents?) were presented. See: Corwin, D.L., and S.M. Lesch. 2013. Protocols and guidelines for field-scale measurement of soil salinity distribution with ECa-directed soil sampling. J. Environ. Eng. Geophysics 18(1):1-25. and: Corwin, D.L., and S.M. Lesch. 2005b. Characterizing soil spatial variability with apparent soil electrical conductivity: I. Survey protocols. Comput. Electron. Agric. 46(1-3):103-134.

Authors responded:

Reply 1.9: We thank the reviewer for highlighting this important issue. For the same site, Jadoon et al. (2015) reported a relationship to relate bulk electrical conductivity to the soil salinity (i.e., the conductivity of the saturated paste extract). Observed soil salinity range between 3-185 dS/m). As discussed earlier (Reply 1.1), in the revised manuscript same relationship will be used to estimate the soil salinity. Text and Figure C5 HESSD Interactive comment Printer-friendly version Discussion paper 9 will be incorporated to show the soil salinity distribution. Synthetic and field measurements were analysed to test the performance of the electromagnetic forward model in conductive and non-conductive soil, and retrieve soil salinity using Bayesian inversion. In the case of synthetic scenarios, EMI data was generated using electromagnetic forward model and Bayesian inversion was used to estimate five parameters (three layer electrical conductivities and two layer thicknesses). Result shows that the electromagnetic forward model is not sensitive to the non-conductive soil. Similarly, Minsley (2011) used synthetic data considering the characteristics of shallow ground-based EMI system, geophex GEM-2 and reported that the electromagnetic forward model is less sensitive to the non-conductive soil. Indeed, in the agriculture field the soil electrical conductivity decreases if the soil water content is below

50% of the field capacity, which may cause the less encouraging results for the nonconductive soils. This issue will be highlighted in the manuscript and the references of Corwin, D.L., and S.M. Lesch. 2005 and 2013 will be incorporate.

While I am satisfied by this reply, I think I might have missed where this issue was addressed in the revised manuscript.

Reply 1.2:

Text had already been incorporated in the last paragraph of results and discussions (278-285 line) and Figure 9 was included to show distribution of soil salinity. References mentioned in the comment were reported in the introduction of the manuscript.

1.3: Congrats for the very interesting research!

Reply 1.3:

We cordially appreciate your critical comments and the time you invested to review our manuscript. Thank you for your compliments.